# Case-Control Analysis of the Impact of Anemia on Quality of Life in Patients with Cancer: A Qca Study Analysis

**DOI:** 10.3390/cancers13112517

**Published:** 2021-05-21

**Authors:** Maria Barca-Hernando, Andres J. Muñoz-Martin, Eduardo Rios-Herranz, Ignacio Garcia-Escobar, Carmen Beato, Carme Font, Estefania Oncala-Sibajas, Alfonso Revuelta-Rodriguez, Maria Carmen Areses, Victor Rivas-Jimenez, Aitor Ballaz-Quincoces, Maria Angeles Moreno-Santos, Juan-Bosco Lopez-Saez, Iria Gallego-Gallego, Teresa Elias-Hernandez, Maria Isabel Asensio-Cruz, Leyre Chasco-Eguilaz, Gonzalo Garcia-Gonzalez, Purificacion Estevez-Garcia, Lucia Marin-Barrera, Remedios Otero-Candelera, Sergio Lopez-Ruz, Jorge Lima-Alvarez, Jose Maria Sanchez-Diaz, Macarena Real-Dominguez, Maria Carmen Borrego-Delgado, Samira Marin-Romero, Luis Jara-Palomares

**Affiliations:** 1Medical Surgical Unit of Respiratory Diseases, Respiratory Department, Hospital Virgen del Rocio, CIBERES, 41013 Sevilla, Spain; maria.barca.sspa@juntadeandalucia.es (M.B.-H.); teresaelias@telefonica.net (T.E.-H.); isabel.asensio.cruz.sspa@juntadeandalucia.es (M.I.A.-C.); lucia.marin.sspa@juntadeandalucia.es (L.M.-B.); rotero@separ.es (R.O.-C.); sergio.lopez.ruz.sspa@juntadeandalucia.es (S.L.-R.); carmen.borrego.sspa@juntadeandalucia.es (M.C.B.-D.); samira.marin.sspa@juntadeandalucia.es (S.M.-R.); 2Medical Oncology Department, Hospital General Universitario Gregorio Marañón, 28009 Madrid, Spain; ajmunoz.hgugm@salud.madrid.org (A.J.M.-M.); igallego.hgugm@salud.madrid.org (I.G.-G.); ggarciag.hgugm@salud.madrid.org (G.G.-G.); 3Hematology and Hemotherapy Department, Hospital Universitario Virgen de Valme, 41701 Sevilla, Spain; eduardo.rios.sspa@juntadeandalucia.es; 4Medical Oncology Department, Hospital Universitario Virgen de las Nieves, 18014 Granada, Spain; ignacio.garcia.escobar.sspa@juntadeandalucia.es; 5Medical Oncology Department, Hospital Virgen de la Macarena, 41009 Sevilla, Spain; mariac.beato.sspa@juntadeandalucia.es; 6Medical Oncology Department, DIBAPS/Translational Genomics and Targeted Therapeutics in Solid Tumors, Hospital Clínic, 08036 Barcelona, Spain; cfont@clinic.cat; 7Emergency Department, Hospital Virgen Macarena, 41009 Sevilla, Spain; estefania.oncala.sspa@juntadeandalucia.es; 8Medical Oncology Department, Hospital Universitario Central de Asturias, 33611 Oviedo, Spain; arevuelta.gea4@sespa.es; 9Medical Oncology Department, Hospital de Orense, 32616 Orense, Spain; maria.carmen.areses.manrique@sergas.es; 10Medical Oncology Department, Hospital de Jerez de la Frontera, 11407 Cádiz, Spain; victorj.rivas.sspa@juntadeandalucia.es; 11Respiratory Department, Hospital de Galdakao-Usansolo, 48960 Bizkaia, Spain; aitor.ballazquincoces@osakidetza.net (A.B.-Q.); leyre.chascoeguilaz@osakidetza.eus (L.C.-E.); 12Medical Oncology Department, Hospital Universitario Puerto Real, 11510 Cádiz, Spain; angeles.moreno@uca.es; 13Internal Medicine Unit, Hospital Universitario, Puerto Real, 11510 Cádiz, Spain; juanbosco.lopez@uca.es; 14Medical Oncology Department, Hospital Virgen del Rocio, 41013 Sevilla, Spain; purificacion.estevez.sspa@juntadeandalucia.es; 15Respiratory Department, Hospital Universitario Virgen de Valme, 41701 Sevilla, Spain; jorge.lima.sspa@juntadeandalucia.es; 16Medical Surgical Unit of Respiratory Diseases, Pharmacy, Respiratory Department, Hospital Virgen del Rocio, 41013 Sevilla, Spain; jmaria.sanchez.diaz.sspa@juntadeandalucia.es; 17Department of Preventive Medicine and Public Health, Universidad de Málaga, 29071 Málaga, Spain; macarenareal@uma.es

**Keywords:** anaemia, neoplasm, quality of life

## Abstract

**Simple Summary:**

The impact of anemia on the quality of life (QoL) in cancer patients has been studied previously; however, the cut-off point used to define anemia differed among studies, thus providing inconsistent results. Therefore, we analysed the clinical impact of anemia on QoL using the same cut-off point for hemoglobin level to define anemia as that used in ESMO clinical practice guidelines. This post-hoc analysis aimed to determine the impact of anemia on QoL in cancer patients through the European Organization for Research and Treatment of Cancer Quality of life questionnaire version 3.0 (EORTC QLQ-C30) and Euro QoL 5-dimension 3-level (EQ–5D–3L) questionnaire. We found that cancer patients with anemia had significantly worse QoL in clinical terms. In addition, anemic patients had more pronounced symptoms than those in non-anemic patients.

**Abstract:**

Anemia is a common condition in cancer patients and is associated with a wide variety of symptoms that impair quality of life (QoL). However, exactly how anemia affects QoL in cancer patients is unclear because of the inconsistencies in its definition in previous reports. We aimed to examine the clinical impact of anemia on the QoL of cancer patients using specific questionnaires. We performed a post-hoc analysis of a multicenter, prospective, case-control study. We included patients with cancer with (cases) or without (controls) anemia. Participants completed the European Organization for Research and Treatment of Cancer Quality of Life questionnaire version 3.0 (EORTC QLQ-C30) and Euro QoL 5-dimension 3-level (EQ–5D–3L) questionnaire. Statistically significant and clinically relevant differences in the global health status were examined. From 2015 to 2018, 365 patients were included (90 cases and 275 controls). We found minimally important differences in global health status according to the EORTC QLQ-C30 questionnaire (case vs. controls: 45.6 vs. 58%, respectively; mean difference: −12.4, *p* < 0.001). Regarding symptoms, cancer patients with anemia had more pronounced symptoms in six out of nine scales in comparison with those without anemia. In conclusion, cancer patients with anemia had a worse QoL both clinically and statistically.

## 1. Introduction

Anemia is a global public health problem with major consequences for human health, as well as for social and economic development [1,2,3]. In the general population, the incidence of anemia is approximately 24.8%, affecting 1.62 billion people [4]. Anemia is common in patients with cancer, particularly among those treated with myelosuppressive chemotherapy. Estimated prevalence of anemia in cancer patients based on epidemiological data analysis varies between 30 and 90% [5]. This variability is explained by the different definitions of anemia, types of malignancy, and the stage of disease [6]. Anemia may develop as a result of the malignant disease process itself, as a consequence of treatment, bleeding, nutritional deficiencies, bone marrow damage, tumor infiltration into the bone marrow, or immunological impairment of erythropoietic response [5,6,7,8].

Hemoglobin level is a quantitative variable used for the diagnosis of anemia, and its cut-off point differs depending on the studies reviewed [9,10,11,12]. The World Health Organization defines the cut-off hemoglobin level for anemia as <12.0 g/dL in women and <13.0 g/dL in men [9]. Some studies that investigated the relationship between hemoglobin level and quality of life (QoL) in anemic cancer patients diagnosed anemia in those with hemoglobin levels <11.0 g/dL [10,11]. In addition, the European Society for Medical Oncology (ESMO) Clinical Practice Guideline established that cancer patients undergoing chemotherapy treatment who had hemoglobin level <11 g/dL and iron deficiency should be administered iron intravenously to correct for this deficiency [12].

Health-related QoL can be defined as a sense of well-being, including physical, emotional, and social dimensions. The use of QoL evaluation has increased significantly in recent decades [13,14]. Different questionnaires are available for the specific assessment of QoL in patients with cancer. The European Organization for Research and Treatment of Cancer Quality of Life Questionnaire version 3.0 (EORTC QLQ-C30) is a reliable questionnaire that has been used to measure the QoL in patients with cancer [15,16,17]. The impact of anemia on QoL has been studied in different populations, including cancer patients, those with chronic obstructive pulmonary disease, heart failure, and chronic kidney disease [18,19,20]. Anemia impairs QoL by causing a variety of symptoms, such as fatigue, tachycardia, respiratory disorders, and cognitive impairment [21]. Demetri et al. reported that increasing hemoglobin levels significantly improved QoL parameters in patients with cancer undergoing chemotherapy and being treated with recombinant human erythropoietin [22].

The impact of anemia on QoL in cancer patients has been previously investigated; however, these studies used different definitions of anemia, thus providing inconsistent results [15,16,17,21,23,24,25,26,27,28]. Therefore, we analyzed the clinical impact on QoL using the same cut-off point for hemoglobin level to define anemia as that used in ESMO clinical practice guidelines; this issue was investigated in a heterogeneous population with different tumor sites and using a control population with similar characteristics. To this end, we performed a post-hoc analysis of the Quality of Life in Cancer Study (QCa study), which was a multicenter, prospective, case-control investigation conducted in 13 Spanish hospitals between June 2015 and January 2018 [29].

## 2. Results

We obtained data on hemoglobin levels in 86% (365/425) of patients included in the study (90 cases and 275 controls). Appendix A shows recruitment per center. The mean age of enrolled participants was 61.4 ± 15.0 years, and 56% were male. The most common comorbidities were hypertension (37.6%) and dyslipidemia (24.2%). One-third (32.3%) of the patients had acute symptomatic VTE. All participants had a histologically confirmed diagnosis of cancer, and the most common types were gastrointestinal (25.2%) and lung (20.8%) cancers. Metastases were identified in 67.8% of the patients, and 77.5% were undergoing oncological treatment. The Eastern Cooperative Oncology Group (ECOG) performance status was 0–1 (90%). Baseline patient characteristics are shown in Table 1. The marital status, education status, employment status, and gross income of the study participants are detailed in Appendix A.

### 2.1. Impact of Anemia on Quality of Life in Patients with Cancer

Of the 365 patients included, 97.2% completed the EQ−5D questionnaire (*n* = 355) and 97% completed the EORTC QLQ-C30 questionnaire (*n* = 354). When global health status was evaluated using the EORTC QLQ-C30, patients with anemia showed clinically and statistically significant differences, showing lower QoL than non-anemic cancer patients (45.6 vs. 58, respectively; mean difference: −12.4, *p* < 0.001) (Table 2). These differences were clinically and statistically significant even after adjustment for the presence or absence of metastases (43.4 vs. 55.3, respectively; mean difference: −11.9, *p* < 0.001). Furthermore, patients with anemia had worse physical functioning (67.2 vs. 78.9, respectively; mean difference: −11.7, *p* < 0.001), role functioning (59.4 vs. 71.5, respectively; mean difference: −12, *p* < 0.05) and emotional functioning (62.7 vs. 72.2, respectively; mean difference: −9.8, *p* < 0.05). Furthermore, anemic patients had more pronounced symptoms (clinically and statistically) in 6 out of 9 scales compared with those in non-anemic patients. The affected scales were fatigue (48.0 vs. 35, respectively; mean difference: +13, *p* < 0.001), dyspnea (21.6 vs. 14.6, respectively; mean difference: +7.0, *p* < 0.05), pain (35.5 vs. 23.8, respectively; mean difference: +11.7, *p* < 0.001), appetite loss (35.7 vs. 21.5, respectively; mean difference: +14.2, *p* < 0.001), nausea and vomiting (20.6 vs. 9.6, respectively; mean difference: +11.0, *p* < 0.001), and diarrhea (21.5 vs. 14.4, respectively; mean difference: +7.1, *p* < 0.05). However, there were no differences between patients with or without anemia when using the EQ–5D–3L questionnaire. Figure 1 shows two sections of the violin plot, indicating differences between cases and controls in the kernel distribution.

### 2.2. Subgroup Analysis

A stratified analysis was performed according to the sex and location of the tumor. According the EORTC QLQ-C30 questionnaire, the presence of anemia was found to affect global health status (with the difference being larger than MID) in men but not in women (42.5 vs. 59.6%, respectively; mean difference: −17.1, *p* < 0.001). Only physical functioning and role functioning were found to be affected in women with anemia. Regardless of sex, we identified significant differences in the scales related to symptoms with higher scores in terms of fatigue (49.0 vs. 33.2 in men, respectively; mean difference: 15.8; 48.3 vs. 37.5 in women, respectively; mean difference: 10.8), pain (19.5 vs. 8.8%, in men respectively; mean difference: 10.7; 21.7 vs. 10.6 in women, respectively; mean difference: 11.1), and nausea and vomiting (35.8 vs. 22.8 in men, respectively; mean difference: 12.2; 36.0 vs. 25.3 in women, respectively; mean difference: 10.7). Conversely, the EQ–5D–3L questionnaire showed no significant differences between the sexes (Appendix A).

According to tumor location, the patients with gynecologic, gastrointestinal, and lung cancers had statistically and clinically significant differences (larger than MID) in functional scales of the EORTC QLQ-C30 questionnaire (Appendix A). In addition, patients with gastrointestinal cancer showed significant differences in fatigue, dyspnea, pain, appetite loss, nausea, and vomiting. Appendix A shows an analysis of symptoms based on the EORTC QLQ-C30 questionnaire, and the results revealed significant differences (larger than the MID) in patients with gynecologic and lung cancer (Appendix A).

We then analyzed the effect of acute symptomatic VTE on the QoL in patients with anemia. The functional scales of the EORTC QLO-C30 questionnaire showed significant differences in physical and role functioning in patients with VTE compared with those without VTE (60.2 vs. 72.3, respectively; mean difference: −12.1, *p* < 0.05), (49.5 vs. 66.6, respectively; mean difference: −17.1, *p* < 0.05). With respect to symptoms, anemic cancer patients without VTE scored higher in appetite loss (43.5 vs. 24.5%, respectively; mean difference: 19.0; Appendix A).

## 3. Discussion

This post-hoc analysis found that the presence of anemia in patients with cancer had a negative impact on QoL. These results were obtained using the same anemia definition proposed by the ESMO clinical practice guidelines; additionally, the study cohort consisted of patients with different tumor locations. The EORTC QLQ-C30 questionnaire showed that anemia was associated with lower global health status, physical functioning, role functioning, emotional functioning, as well as higher fatigue, dyspnea, pain, appetite loss, diarrhea, nausea, and vomiting. A previous study that assessed the impact of anemia on the QoL in 80 patients with cancer who completed EORTC-QLQ−30 questionnaire supported these findings, although in this work anemia was defined using hemoglobin level <6.5 g/dL [15]. Nevertheless, that was a study with a small sample size that represented a younger population (76.2% were younger than 60 years old), with a predominance of gynecologic cancer (51.2%) and more severe anemia. Cella et al. developed the Functional Assessment of Cancer Therapy-Anemia (FACT-An) to evaluate the impact of anemia on cancer patients. The findings in these studies showed that patients with hemoglobin levels >12 g/dL reported significantly less fatigue, better physical and functional well-being, and higher overall QoL than those with hemoglobin levels <12 g/dL [21]. In addition, a prospective multicenter study performed in Japan enrolled 227 cancer patients to analyze QoL using the FACT-An. Patients with cancer and hemoglobin <11 g/dL had significant lower FACT-An scores than those with hemoglobin ≥11 g/dL) (120.2 ± 24.9 vs. 130.5 ± 24.3; *p* = 0.004) [24]. However, a majority of the patients had lung cancer and were administered platinum-based chemotherapy, which may have aggravated anemia and affected QoL. Wedding et al. performed a prospective study to evaluate the association between anemia, QoL (using the EORTC-QLQ-C30 questionnaire), and functional status (Karnofsky Performance Status KPS) in 447 patients [17]. The authors found that older patients (cancer or non-cancer) had worse QoL, but anemia did not have an influence on QoL in younger patients with cancer. These findings may be due to the fact that all patients included were hospitalized and had worse general conditions than those enrolled in an outpatients setting.

Several placebo-controlled and open-label studies have demonstrated improvements in QoL through the treatment of anemia with erythropoiesis-stimulating proteins in patients with cancer [30,31]. In a recent randomized placebo-controlled trial that included 375 patients with non-myeloid hematologic malignancies, patients who were treated with epoetin alfa had a significant increase in hemoglobin levels and reported significant improvement in overall QoL (epoetin alfa + 4.8; placebo −6.0; *p* = 0.0048), energy levels (epoetin alfa 8.1; placebo −5.8; *p* = 0.007), and ability to perform daily activities (epoetin alfa +7.5; placebo −6.0; *p* = 0.0018) [30]. Hudis et al. performed an open-label, non-randomized, multicenter study to evaluate the effects of epoetin alfa in patients with stage I-III breast cancer and baseline hemoglobin levels of ≥10 and ≤14 g/dL. Post-hoc analyses showed that patients who had baseline hemoglobin ≥10 to ≤12 g/dL had significant improvements in energy, activity, and overall QoL scores after 12 weeks of epoetin alfa therapy at the final measurement (*p* < 0.005) [31]. A recent study suggested that hemoglobin levels >12 g/dL during erythropoietic therapy have been associated with an increased risk of thrombotic vascular events; therefore, these treatments are not recommended in cancer patients with hemoglobin levels >12 g/dL [32].

This study had several strengths. First, although the impact of anemia on the QoL in cancer patients has been studied previously, it is important to know the impact of anemia considering the cut-off point indicated in the clinical practice guidelines [12], in a heterogeneous population with different tumor sites and using a control population with similar characteristics that allowed us to obtain direct comparisons of QoL, without resorting to historical cohorts characterized by different demographic characteristics. According to the ESMO clinical practice guidelines, which recommend that cancer patients undergoing chemotherapy treatment and with hemoglobin level <11 g/dL should receive iron treatment, we established the variable anemia at a hemoglobin level <11 g/dL. Second, our QoL analysis considered minimally important differences, being able to identify clinically relevant differences that were not only statistically significant without clinical benefit. Third, the EORTC C−30 questionnaire was found to be a useful and appropriate tool for analyzing the QoL in patients with anemia.

Our study also had some limitations. First, it was a post-hoc analysis of a case-control study, and all variables could not be matched. Nevertheless, we recruited 365 patients with a case:control ratio of 1:3. Second, this study was not initially designed to analyze the impact of anemia on QoL in cancer patients, and for that reason, other specific tools were not used to adequately assess anemia. Therefore, in future studies, it would be interesting to include scales such as FACT-An, which was developed specifically to assess the impact of anemia in cancer patients [21]. Other tools frequently used in previous studies of patients with anemia included the LASA and the Functional Assessment of Cancer Therapy-Fatigue (FACT-F) scale [10,31]. The use of these scales (FACT-An and FACT-F) would have allowed us to obtain additional information, ultimately representing a caveat inherent in our study. Moreover, other variables that could affect QoL, such as iron levels, erythropoietin treatment, blood transfusions, or recent hospitalizations were not included. It would have been interesting to know the impact on QoL taking these variables into account; unfortunately, we did not have these data. In this scenario, subgroup analyses would require a larger sample size to ensure a sufficient statistical power. Finally, we did not find differences in the generic questionnaire, such as EQ–5D–3L. This may be due to low statistical power that could be solved by increasing recruitment in future studies.

## 4. Materials and Methods

### 4.1. Study Design

This was a post-hoc analysis of QCa. Patients included were divided into two groups: cases (patients with anemia) and controls (patients without anemia). As previously suggested by the ESMO Clinical Practice Guideline [12] and other studies on cancer patients [10,11], anemia was defined as a condition with hemoglobin level <11 g/dL. Several variable related to anemia were analyzed in this study, including the presence of metastasis, tumor location, and the use of chemotherapy. In this regard, the presence of distant metastases from solid tumors may invade the bone marrow and disrupt erythropoiesis, resulting in anemia. Additionally, chemotherapy may cause anemia through a number of different mechanisms, including stem cell death, oxidant damage to mature hematopoietic cells, and immune-mediated hematopoietic cell destruction. Finally, tumor localization may be relevant to the development of anemia in relation to the risk of acute or chronic blood loss, as in gastrointestinal, genitourinary and gynecologic malignancies [33]. The complete protocol has been previously published in detail [29,34]. A total of four QoL questionnaires were evaluated among the included patients, showing that acute symptomatic VTE adversely affected their QoL. In patients with symptomatic VTE, all questionnaires were completed within 30 days of the index thrombotic event.

All questionnaires were analyzed anonymously and completed simultaneously to avoid confounding factors. The Research Ethics Committee of the Virgen del Rocio Hospital, Sevilla, Spain (code: 0191-N-14) reviewed and approved the study protocol. All of the eligible patients provided written informed consent.

The objective of this post-hoc analysis was to evaluate the impact of anemia on the QoL in patients with cancer. With this aim, we compared the global health status between patients in cases and controls groups using two QoL questionnaires: (1) EORTC QLQ-C30 version 3.0 [35], which is a specific questionnaire for patients with cancer; (2) EQ-5D-3L [36], which is a generic questionnaire to evaluate QoL. The EORTC QLQ-C30 is a reliable and valid system that is composed of five functioning scales, three symptom scales, one global health scale, and six independent items. High scores on the global health scale and functioning status reflect a better QoL, and high scores on the symptom scale indicate a reduction in QoL [37,38]. The minimally important difference (MID) on the EORTC QLQ-C30 questionnaire varied from 6 to 15 points [39,40,41]. The EQ–5D–3L questionnaire has been widely used in cancer and anemia studies [42,43]. The questionnaire elicits information on five aspects of QoL which are scored from 1 (no problems) to 3 (unable to/extreme problems) [44]. Responses to the five EQ–5D items define a health state for which an index score can be generated. The index score was anchored at 0 (death) and 1 (full health), with higher scores corresponding to higher health states (scores ranging from −0.654 to 1). The MID on the EQ–5D questionnaire for patients with malignancies was 0.06–0.08 [45].

### 4.2. Statistical Analysis

Descriptive statistics are expressed as absolute or relative frequencies (categorical variables) and means and standard deviations (continuous variables). Continuous data with an asymmetric distribution are presented as medians and interquartile ranges. The mean, median and percentage of respondents—along with the minimum/maximum scores—are presented for each questionnaire. For this post-hoc analysis, we calculated the effect size (Cohen’s d) and the relative efficiency (RE) to examine the questionnaire performance. The effect size was obtained using the difference in mean scores divided by the pooled standard deviation. The effect sizes can vary from mild (0.2) to moderate (0.5) or large (0.8). The RE statistic was calculated as the ratio of F-statistics in the analysis of variance (ANOVA) tests of the differences in scores between known groups. A greater efficiency or discrimination is related to higher F-statistic values. Figure 1 shows a violin plot. In general, violin plots are a method of plotting numeric data and can be considered as a combination of a box plot with a kernel density plot. In the violin plot, the same information as in the box plot can be identified, as follows: median value (black line inside the box) and interquartile range (edges of the box). The Kernel density estimation (KDE) is a non-parametric approach for estimating the probability density function of a random variable. In addition, these plots depict the actual distribution of the observed values (dots in cases and triangles in controls). The kernel density plot (grey line) represents the probability density function. A higher probability of observation at any given value is related to larger sections of the violin plot.

## 5. Conclusions

In summary, the results of our study demonstrate that the presence of anemia has an adverse impact on the QoL in patients with cancer. Therefore, it is crucial to offer appropriate guidelines for the management and treatment of these patients.

## Figures and Tables

**Figure 1 cancers-13-02517-f001:**
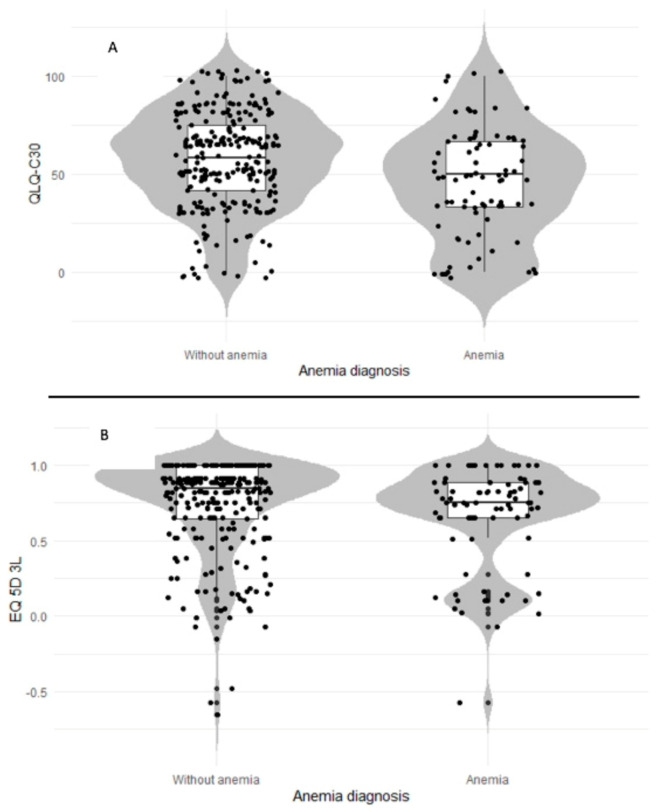
Patient data distribution and probability densities within different questionnaires. Patients with anemia are indicated as “cases” and those without anemia as “controls”. Violin plots for QLQ-c30 Global Health Status (**A**) and EQ–5D–3L Index (**B**). The plots indicate the distribution of the observed values (dots). The kernel density plot (grey line) represents the probability density function. Larger sections of the violin plot indicate a higher probability of observation at a given value, whereas thinner sections correspond to a lower probability.

**Table 1 cancers-13-02517-t001:** Clinical characteristics of the study patients.

Variable	Anemia Present (Cases)	Anemia Absent (Controls)	Entire Cohort
Male sex, *n* (%)	43 (47.8%)	161 (58.5%)	204 (55.9%)
Age, years (*n* = 365); mean ± SD	63.2 (11.8)	60.8 (15.9)	61.4 (15)
Body mass index, kg/m^2^ (*n* = 312), mean ± SD	26.2 (4.9)	26.4 (5.3)	26.4 (5.1)
Arterial hypertension (*n* = 364), *n* (%)	42 (46.7%)	95 (34.7%)	137 (37.6%)
Dyslipidemia (*n* = 363), *n* (%)	23 (25.8%)	65 (23.7%)	88 (24.2%)
Diabetes mellitus (*n* = 364), *n* (%)	16 (17.8%)	42 (15.3%)	58 (15.9%)
Asthma (*n* = 364), *n* (%)	5 (5.6%)	2 (0.7%)	7 (1.9%)
Acute coronary syndrome (*n* = 364), *n* (%)	4 (4.4%)	5 (1.8%)	9 (2.5%)
Stroke (*n* = 364), *n* (%)	5 (5.6%)	8 (2.9%)	13 (3.6%)
Chronic kidney disease (*n* = 364), *n* (%)	3 (3.3%)	9 (3.3.%)	12 (3.3%)
Smoking (*n* = 364), *n* (%)	20 (22.2%)	68 (24.8%)	88 (24.2%)
VTE (*n* = 365), *n* (%)	39 (43.3%)	79 (28.7%)	118 (32.3%)
Non-steroidal anti-inflammatory drugs (*n* = 364), *n* (%)	9 (10%)	23 (8.4%)	32 (8.8%)
Statins (*n* = 364), *n* (%)	16 (17.8%)	35 (12.8%)	51 (14%)
Active anticancer treatment, *n* (%)	71 (79.8%)	208 (76.8%)	279 (77.5%)
Central venous catheter, *n* (%)	13 (14.8%)	62 (22.8%)	75 (20.8%)
ECOG performance status (*n* = 339)			
0, *n* (%)	16 (19%)	109 (42.7%)	125 (36.9%)
1, *n* (%)	54 (64.3%)	128 (50.2%)	182 (53.7%)
2, *n* (%)	13 (15.5%)	13 (5.1%)	26 (7.7%)
3, *n* (%)	1 (1.2%)	4 (1.6%)	5 (1.5%)
4, *n* (%)	0 (0%)	1 (0.4%)	1 (0.3%)
Metastasis (*n* =345), *n* (%)	58 (68.2%)	176 (67.7%)	234 (67.8%)
Tumor site (*n* = 365)			
Gynecologic, *n* (%)	20 (22.2%)	22 (8.0%)	42 (12)
Lung, *n* (%)	17 (18.9%)	59 (21.5%)	76 (20.8%)
Digestive, *n* (%)	20 (22.2%)	72 (26.2%)	92 (25.2%)
Genitourinary, *n* (%)	8 (8.9%)	19 (6.9%)	27 (7.4%)
Lymphoma, *n* (%)	9 (10%)	43 (15.6%)	52 (14.2%)
Other sites, *n* (%)	16 (17.8%)	60 (21.8%)	76 (20.8%)

Abbreviations: SD: standard deviation; ECOG: Eastern Cooperative Oncology Group.

**Table 2 cancers-13-02517-t002:** Differences in EORTC QLQ-C30 functional and symptoms scales and EQ–5D–3L between cancer patients with and without anemia.

EORTC QLQ-C30 Functional Scale ^1^	Cases (Cancer with Anemia)	Controls (Cancer without Anemia)	Mean Difference	Effect Size (95% CI)	Relative Efficiency (95% CI)
Global health status	45.6	58	−12.4 **	0.78 (0.54; 1.1)	−0.48 (−0.23; −0.73)
Physical functioning	67.2	78.9	−11.7 **	0.71 (0.49; 1)	−0.50 (−0.25; −0.75)
Role functioning	59.4	71.5	−12 *	0.86 (0.59; 1.2)	−0.37 (−0.12; −0.62)
Emotional functioning	62.7	72.2	−9.8 *	0.67 * (0.46; 0.94)	−0.38 (−0.13; −0.63)
Cognitive functioning	80.7	84.1	−3.4	0.86 (0.6; 1.2)	−0.14 (−0.39; 0.1)
Social functioning	64.2	72.8	−8.3	0.75 (0.52; 1.05)	−0.27 (−0.02; −0.52)
EORTC QLQ-C30 symptoms scale ^2^	Cases (Cancer with anemia)	Controls (Cancer without anemia)	Mean difference	Effect size (95% CI)	Relative efficiency (95% CI)
Fatigue	48	35	13 **	0.91 (0.3; 1.3)	0.48 (0.2; 0.7)
Nausea and vomiting	20.6	9.6	11 **	0.34 ** (0.2; 0.5)	0.52 (0.3; 0.7)
Pain	35.5	23.8	11.7 **	0.8 (0.5; 1.1.)	0.41 (0.1; 0.6)
Dyspnea	21.6	14.6	7 *	0.75 (0.5; 1)	0.25 (0.1; 0.5)
Insomnia	34.1	30.1	4	0.9 (0.6; 1.3)	0.12 (0.1; 0.4)
Appetite loss	35.7	21.5	14.2 **	0.6 * (0.4; 0.9)	0.45 (0.2; 0.7)
Constipation	26.6	22.3	4.3	0.87 (0.6; 1.2)	0.14 (0.1; 0.4)
Diarrhea	21.5	14.4	7.1 *	0.57 * (0.4; 0.8)	0.26 (0.01; 0.5)
Financial difficulties	17.4	19.9	−2.5	1.12 (0.7; 1.5)	−0.08 (−0.3; 0.1)
EQ–5D–3L ^3^	Cases (Cancer with anemia)	Controls (Cancer without anemia)	Mean difference	Effect size (95% CI)	Relative efficiency (95% CI)
Index score	0.65	0.73	−0.08	0.88 (0.6; 1.2)	−0.24 (−0.4; 0)

Abbreviations: CI: confidence interval. * *p* < 0.05; ** *p* < 0.001; ^1^ EORTC QLQ-C30 functional scale: Higher scores reflect a better health status and the minimally important difference varies from 6 to 15 points; ^2^ EORTC QLQ-C30 symptoms scale: Higher scores in the symptoms scale indicate a reduction in quality of life; ^3^ EQ–5D–3L: Higher scores reflect a better health status and the minimally important difference on the EQ–5D questionnaire for patients with malignancies is 0.06–0.08 UI.

## Data Availability

For the original data set, please contact luisj.jara.sspa@juntadeandalucia.es. Individual participant data were not shared.

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
