# Peer review of "Case-Control Analysis of the Impact of Anemia on Quality of Life in Patients with Cancer: A Qca Study Analysis"

_cancers, 2021, doi:10.3390/cancers13112517_

Round 1
Reviewer 1 Report
Thank you for the opportunity to review this paper. Overall I found it focused and what is written, clear. However I think there are several changes that could be made to strengthen your paper.
- I was missing a clear description about the rationale for your study at the onset and what makes this manuscript/study unique. It is needed to set the stage for the work clearly and to help the reader know why it is important to do this study. What makes it unique and what will it add to the literature that we do not know. I found this type of information, finally, at line 226-230 - this part of the presentation should be moved to the early part of the manuscript.
- In the beginning it is important for your reader to know why you have selected certain variables to study - and why you think they would be important in relation to anemia. This needs to be stated clearly and not left for the reader to assume.
- Also, I struggled with having the method and analysis toward the end of the end of the manuscript. The methods and analysis are critically important for understanding the results as they are presented and for interpreting them as a reader. These need to be moved earlier in the paper after the opening and background and before the presentation of the results.
Reviewer 2 Report
I have read the paper of the authors about the anemia and quality of life issues in cancer patients. While the study has been performed adequately, I would like to ask what these results adds to the current literature. You have mentioned in the discussion that scores are existing for anemia and fatigue. You have done a comparison between anemic and non-anemic patients but the difference was not controlled for other variables , like treatment of anemia, metastasis, blood transfusion, etc. It would be interesting to compare the FACT-An or FACT-F scale with the EQ-5D-3L scores.
